# Effect of Surface Chemistry and Crystallographic Parameters of TiO$_2$ Anatase Nanocrystals on Photocatalytic Degradation of Bisphenol A

**Gregor Žerjav [1], Albin Pintar [1,]\*, Michael Ferentz [2], Miron Landau [2], Anat Haimovich [3], Amir Goldbourt [3] and Moti Herskowitz [2]**

[1] Department for Inorganic Chemistry and Technology, National Institute of Chemistry, Hajdrihova 19, SI-1001 Ljubljana, Slovenia; gregor.zerjav@ki.si
[2] Blechner Center for Industrial Catalysis and Process Development, Department of Chemical Engineering, Ben-Gurion University of the Negev, Beer Sheva 84105, Israel; Ferenm@gmail.com (M.F.); mlandau@bgu.ac.il (M.L.); herskow@bgu.ac.il (M.H.)
[3] School of Chemistry, Raymond and Beverly Sackler Faculty of Exact Sciences, Tel Aviv University, Ramat Aviv 6997801, Israel; anathaim@post.tau.ac.il (A.H.); amirgo@tauex.tau.ac.il (A.G.)
**\*** Correspondence: albin.pintar@ki.si; Tel.: +386-1-4760-237

**Abstract:** The photocatalytic activity of a series of anatase TiO$_2$ materials with different amounts of exposed (001) facets (i.e., 12% (TiO$_2$-1), 38% (TiO$_2$-3), and 63% (TiO$_2$-3)) was tested in a batch slurry reactor towards liquid-phase bisphenol A (BPA, c$_0$(BPA) = 10 mg/L, c$_{cat.}$ = 125 mg/L) degradation. Photo-electrochemical and photo-luminescence measurements revealed that with the increasing amount of exposed anatase (001) facets, the catalysts generate more electron-hole pairs and OH· radicals that participate in the photocatalytic mineralization of pollutants dissolved in water. In the initial stages of BPA degradation, a correlation between % exposure of (001) facets and catalytic activity was developed, which was in good agreement with the findings of the photo-electrochemical and photo-luminescence measurements. TiO$_2$-1 and TiO$_2$-3 solids achieved 100% BPA removal after 80 min in comparison to the TiO$_2$-2 sample. Adsorption of BPA degradation products onto the TiO$_2$-2 catalyst surface was found to have a detrimental effect on the photocatalytic performance in the last stage of the reaction course. Consequently, the global extent of BPA mineralization decreased with the increasing exposure of anatase (001) facets. The major contribution to the enhanced reactivity of TiO$_2$ anatase (001) surface is the Brønsted acidity resulting from dissociative chemisorption of water on a surface as indicated by FTIR, TPD, and MAS NMR analyses.

**Keywords:** heterogeneous photocatalysis; photocatalytic oxidation; bisphenol A; TiO$_2$; nanoparticles shape; anatase (001) facets

## 1. Introduction

Heterogeneous photocatalytic oxidation of organic pollutants present in an aqueous medium is one of the most promising advanced oxidation processes (AOPs) for waste water treatment. Due to its properties (high photocatalytic activity, chemical stability, low cost, nontoxicity, long-term stability against photo-corrosion, biological inertness, etc.) semiconductor titanium dioxide (TiO$_2$) presents one of the most suitable material to be used in AOPs as a photo-catalyst [1,2]. Photocatalytic activity of a semiconductor is triggered with the illumination of its surface by a photon, the energy of which is equal to or greater than the band gap energy (E$_g$) of the semiconductor. A major drawback of TiO$_2$ is its band gap energy of 3.0 to 3.2 eV, meaning that it can be excited only by ultraviolet (UVA) light (λ < 387 nm). One of the strategies to overcome this drawback is nitrogen doping of TiO$_2$ [3]. This results in the

creation of mid-gap states, which means that the energy of visible light is sufficient to trigger the photocatalytic activity of $TiO_2$ based catalyst. The energy of the photon excites an electron ($e^-$) in the valence band (VB) of the semiconductor and transfers it to the conduction band (CB), leaving a positive hole ($h^+$) in VB. Photo-generated charge carriers may reduce and oxidize the substrates adsorbed on the catalyst surface and produce reactive oxygen species (ROS) needed for degradation of water dissolved pollutants. On the other hand, if there is no chemical reaction, the photo-generated $e^-$ and $h^+$ recombine with each other and cannot participate in oxidative degradation reactions. Fast electron-hole recombination is recognized as the second major drawback of $TiO_2$, which significantly reduces its photocatalytic activity.

In our previous studies [4–6], we used different approaches to overcome the drawbacks of $TiO_2$, thus improving its use as a photo-catalyst in heterogeneous photocatalytic oxidation of organic pollutants dissolved in water. We combined $TiO_2$ with the low band gap semiconductor, $Bi_2O_3$, to obtain a visible-light active photo-catalyst [3], or decorated different types of anatase $TiO_2$ nanorods with reduced graphene oxide (rGO) [4] to slow down the electron-hole recombination. In one type of $TiO_2$ nanorod, more reactive anatase (001) facets were exposed in comparison to the other $TiO_2$ nanorods utilized, where thermodynamically more stable (101) facets prevailed. Subsequently, the $TiO_2$ + rGO composites with (001) facets enabled up to 6-fold higher BPA degradation than composites with (101) facets. Other authors [7–9] substantiated that (001) facets of anatase $TiO_2$ preferentially accumulate photo-generated holes, which upon contact with water generate the highly active hydroxyl radicals (OH·). According to Selloni [10], the origin of the high reactivity of anatase (001) facets is twofold: (i) The high density of surface under-coordinated Ti atoms, and (ii) the very strained configuration of the surface atoms. There are very large Ti-O-Ti bond angles at the surface, meaning that 2p states on the surface oxygen atoms are destabilized and very reactive. The (001) facets are much more reactive in the dissociative chemisorption of water than (101) facets, which normally form the majority of the surface area. Yang et al. [11] prepared anatase $TiO_2$ single-crystal nanosheets exhibiting 64% exposure of (001) facets; the measured production of OH· radicals was found to be five times higher in comparison to OH· production obtained over the benchmark material, P25. Further, a micro-sheet anatase $TiO_2$ single crystal photocatalyst with a remarkable 80% level of reactive (001) facets increased the ability for photocatalytic oxidation of water dissolved 4-chlorophenol under UV irradiation [12]. Observations of Xiang et al. [13] suggest that the photocatalytic selectivity of $TiO_2$ catalyst towards photodecomposition of methyl violet was facilitated by the exposure of reactive (001) facets. High photocatalytic activity of (001) anatase facets for the decomposition of water dissolved dyes was observed by Liu [14]. He et al. reported [15] that due to the exposure of (001) facets, more efficient $CO_2$ reduction was achieved due to improved electron−hole pair separation.

However, as demonstrated by Liu et al. [16], the amount of exposed (001) anatase facets is not the only parameter determining catalytic activity of $TiO_2$ catalysts. For instance, they observed that the anatase $TiO_2$, in the form of 30 to 85 nm truncated octahedra containing 18% (001) facets, exhibited superior photocatalytic activity for the generation of OH· radicals and water splitting compared to the ultrathin anatase $TiO_2$ sheets with 72% exposure of (001) facets. According to [17,18], increasing the exposure of (001) facets in $TiO_2$ nanocrystals decreases their photocatalytic activity in the decomposition of water. For photodegradation of phenol [19] and $CO_2$ methanation [6], increasing the exposure of (001) facets in $TiO_2$ anatase nanocrystals increased the catalytic activity to a maximum value. These differences are strongly dependent on the type of photocatalytic reaction, and are explained by synergistic effects in photocatalysis between (001) and (001) facets of anatase nanocrystals [6,18,20]. They may also be caused by the different chemical functionality of the corresponding crystallographic facets contributing to the surface transformation of substrates, i.e., Lewis and Brønsted acid sites [21–23]. Therefore, the development of efficient $TiO_2$-based catalysts for specific photocatalytic reactions requires careful consideration of materials' crystal morphology and surface chemistry.

Bisphenol A (BPA)—2,2-bis(4-hydroxyphenyl) propane—is a monomer for the production of polycarbonate and epoxy resins in the polymers industry [24]. The leaching of BPA into groundwater

is possible due to the fact that these polymers are used to produce different packaging materials and as such end as waste in landfills. Contamination of the environment by BPA has been a matter of great concern because this compound can damage the endocrine system by working as natural hormones [25,26]. BPA also impairs brain development in fetuses and children [27–29]. Photocatalytic degradation of BPA with $TiO_2$-based catalysts is one of the efficient methods for water detoxification, reducing the risks mentioned above [30–32].

Tailoring the specific atomic configuration and associated surface reactivity of $TiO_2$ nanocrystals by changing the crystallographic parameters (nanocrystals shape) and surface chemistry of $TiO_2$ anatase is one of the approaches that would thus lead to an improved photocatalytic activity of $TiO_2$ and consequently better use of $TiO_2$ as a photo-catalyst in advanced oxidation processes (AOPs) for waste water treatment. Several attempts to improve the photocatalytic activity of $TiO_2$ anatase in the degradation of BPA implementing $TiO_2$ nanocrystals of different shapes—hexagonal microrods [33], mesoporous $TiO_2$ microspheres [34], and hierarchical $TiO_2$ nanoflacks [35]—demonstrated the great potential of this approach. However, an absence of quantitative correlations between the exposure of (001) and (101) crystallographic facets in the primary nanocrystals of these materials and their surface chemistry with their performance in BPA photodegradation limits the understanding of the observed effects.

In this work, a series of anatase $TiO_2$ materials consisting of truncated bipyramid nanocrystals of Wulff construction [36] with crystal size (length/thickness) of 16 to 24 and 5 to 15 nm, surface area of 65 to 150 $m^2/g$, and exposure of (001) facets in the range of 12% to 63% were synthesized and thoroughly characterized by various techniques. The photocatalytic activity of the prepared solids was tested in the liquid-phase BPA oxidation conducted in a batch slurry reactor. The obtained results concerning both the activity and selectivity (i.e., extent of mineralization) of photocatalytic oxidation of aqueous BPA solution were correlated to the morphological, surface, and electronic properties of the synthesized solids, which was the main objective of the present study.

## 2. Results and Discussion

### 2.1. Catalyst Characterization

#### 2.1.1. $TiO_2$ Texture and Crystal Shape

According to the XRD patterns illustrated in Figure 1, all the materials displayed tetragonal structure $TiO_2$—anatase (space group $I4_1/amd$, ICDD Card 21-1272)—having Z = 4 $TiO_2$ units per cell. The HRTEM images (Figure 2) further show that in polycrystalline anatase, the domains of coherent scattering determining the XRD patterns of the $TiO_2$ samples are tetragonal bipyramids (Figure 3) with two base sides, which have a crystal lattice orientation. The eight lateral faces have an orientation (101). The texture characteristics (BET surface area, total pore volume, and average pore size), crystals sizes determining the width/length of these bipyramids, and % exposure of (001) planes in individual nanocrystals and their aggregates are given in Table 1. The % exposures of (001) planes were calculated based on the A and C dimensions of $TiO_2$ nanocrystals (Figure 3) as described in [37]. The A and B dimensions were calculated from XRD data by means of the Scherrer formulation, yielding correct averaging among individual nanocrystals, whose dimensions in distinct samples may vary. It should be noted that the HRTEM micrographs of $TiO_2$ nanocrystals with an appropriate orientation relative to the electronic beam (Figure 2) demonstrate a clear trend of a gradual refining of the nanocrystals' thickness and widening of their length from $TiO_2$-1 to $TiO_2$-3. As a result, the exposure of (001) planes rises in this sample sequence from 12% to 63% of the total surface area, which corresponds to an increase of the surface area related to (001) facets from 18 to 73 $m^2/g$. According to the TEM data, the nanocrystals in the materials, $TiO_2$-1 and $TiO_2$-2, form aggregates with a random orientation of the crystallites. However, this does not affect the % exposure of (001) planes. In the case of the $TiO_2$-3 sample, the nanocrystals form aggregates where crystallites are attached to each other by (001) planes that decrease the exposure of these facets by a factor proportional to the crystals' aggregation ratio [38].

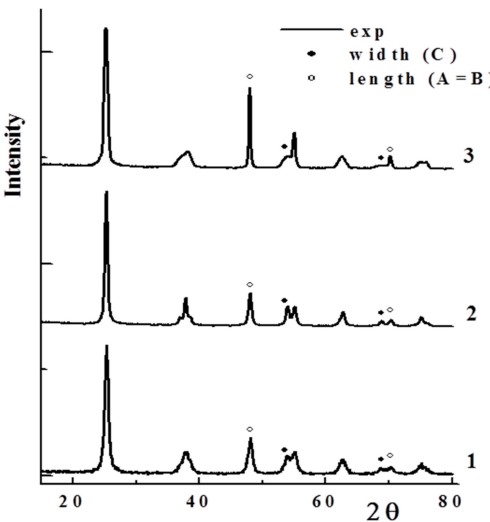

**Figure 1.** XRD patterns of TiO$_2$ materials used in the present study: 1—TiO$_2$-1, 2—TiO$_2$-2, 3—TiO$_2$-3. The XRD peaks used for the calculation of crystals' width and length are denoted by solid and open circles, respectively.

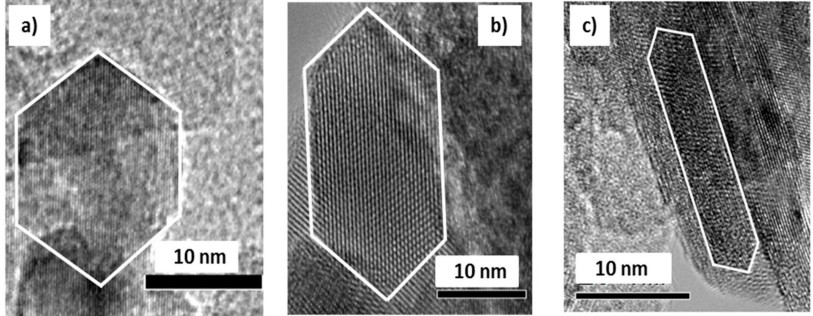

**Figure 2.** HRTEM images of TiO$_2$ materials used in the present study: (**a**) TiO$_2$-1, (**b**) TiO$_2$-2, (**c**) TiO$_2$-3.

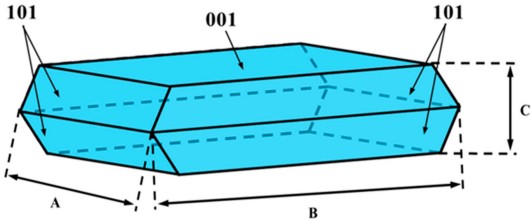

**Figure 3.** Scheme of TiO$_2$ tetragonal bipyramids.

**Table 1.** Texture, nanocrystals' shape characteristics, and % exposure of (001) planes in TiO$_2$ samples used in the present study.

| Sample | Texture Parameters | | | Mean Crystal Size, nm | | | (001) Plane Exposure | |
| | | | | | | | Crystals | Aggregates |
| | Surface Area, m$^2$/g | Total Pore Volume, cm$^3$/g | Average Pore Diameter, nm | Length (A = B) | Thickness C | % | % | m$^2$/g |
|---|---|---|---|---|---|---|---|---|
| TiO$_2$-1 | 150 ± 15 | 0.43 ± 0.005 | 11 ± 0.1 | 16 ± 0.02 | 15 ± 0.02 | 12 ± 0.01 | 12 ± 0.01 | 18 ± 2 |
| TiO$_2$-2 | 65 ± 7 | 0.15 ± 0.002 | 5 ± 0.05 | 24 ± 0.02 | 12 ± 0.01 | 38 ± 0.04 | 38 ± 0.04 | 25 ± 2 |
| TiO$_2$-3 | 120 ± 12 | 0.46 ± 0.005 | 18 ± 0.2 | 24 ± 0.02 | 5 ± 0.005 | 66 ± 0.07 | 63 ± 0.07 | 73 ± 7 |

### 2.1.2. FTIR-ATR, Photo-Electrochemical, and Photo-Luminescence Measurements

The acquired FTIR-ATR spectra of the prepared catalysts are presented in Figure 4. The spectra show strong absorption of IR radiation in the range of 1000–450 cm$^{-1}$, which corresponds to lattice

Ti-O-Ti vibrations. The broad band in the range of 3700–2500 cm$^{-1}$ and a peak at 1640 cm$^{-1}$ can be ascribed to stretching and bending vibrations of surface hydroxyl groups and water, respectively [39]. With the increasing amount of exposed (001) facets in TiO$_2$ nanocrystals, the concentration of terminal Ti-OH hydroxyl groups on the surface increases [13]. This can be seen in Figure 4 as more abundant peaks in the range of 3700–2500 cm$^{-1}$. To further investigate the influence of an increased concentration of surface hydroxyl groups on the behavior of the examined catalysts, the surface acidity and basicity were investigated by performing NH$_3$- and CO$_2$-TPD measurements (Section 2.1.3).

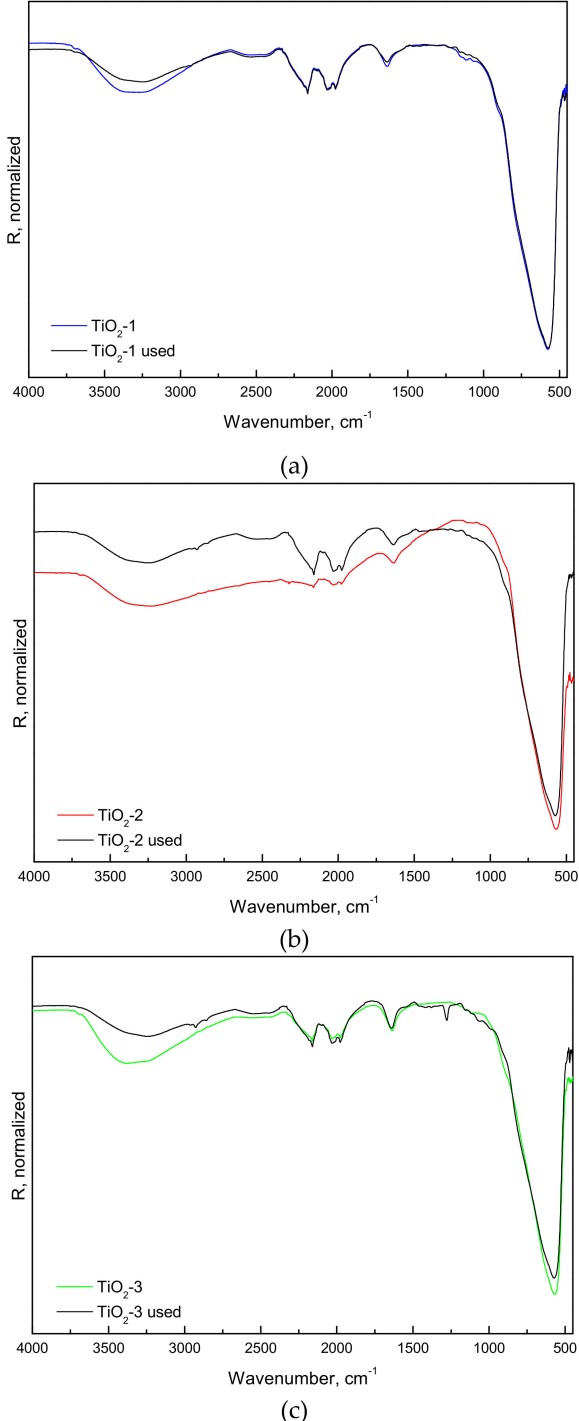

**Figure 4.** FTIR-ATR spectra of (**a**) TiO$_2$-1, (**b**) TiO$_2$-2 and (**c**) TiO$_2$-3 samples before and after use in bisphenol A BPA degradation runs.

No differences among the TiO₂-1 to TiO₂-3 samples were found in the UV-Vis diffuse reflectance spectra (not shown). The light absorption edge occurred in all cases at 385 nm, which corresponds to band gap energy of 3.22 eV.

The separation of the photo-generated electron-hole pairs of all prepared solids was systematically tested by photocurrent measurements (Figure 5). It can be seen that all samples, when illuminated with UVA light, generated anodic photocurrent. Furthermore, TiO₂-1 to TiO₂-3 samples exhibited a stable response in the photocurrent density measurements conducted in six subsequent illumination cycles. The amount of generated current density increased with the increased amount of exposed (001) facets in the TiO₂ solids, since the TiO₂-3 sample had more charge carriers available to produce hydroxyl radicals (OH·) [20,40] compared to the other two samples. Interestingly, the TiO₂-1 sample, although exhibiting the largest BET surface area (Table 1), generated the lowest current density among the solids examined. This implies that surface area is not the governing factor determining their photocatalytic activity. It is interesting to see that in the case of the TiO₂-3 sample, the noise in the corresponding photocurrent curve, after illumination, is more intensive compared to the other two tested samples. The noise may originate from the electron-hole recombination, which, as it is seen in Figure 5, is more intensive for the TiO₂-3 sample. Nevertheless, the highest total amount of photo-generated charge carriers, among all tested samples, was found for the TiO₂-3 sample.

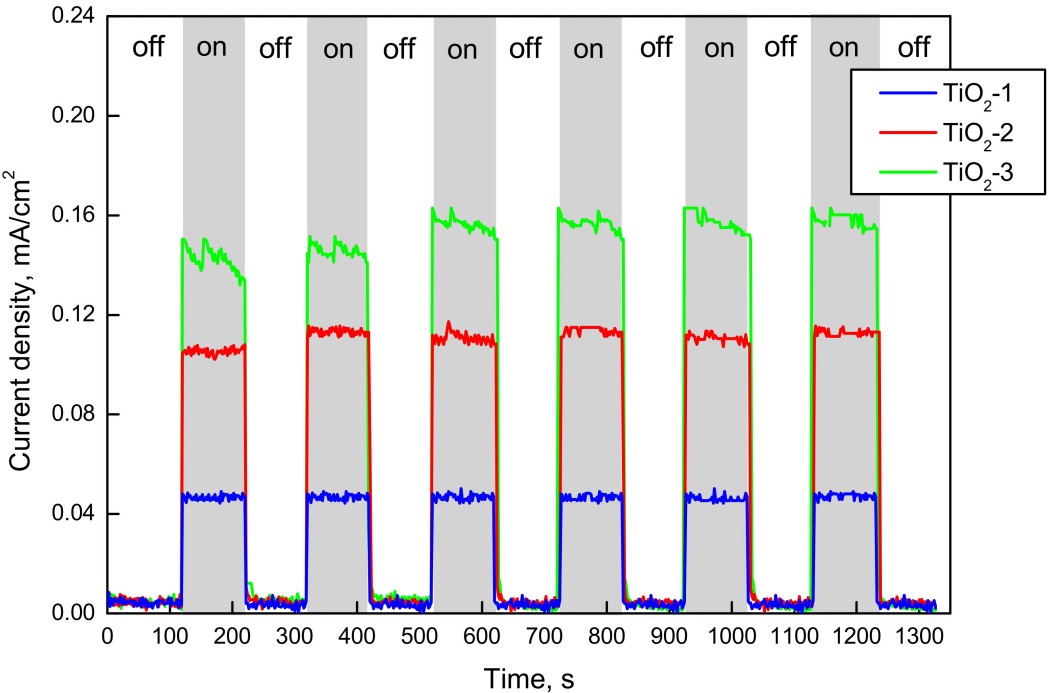

**Figure 5.** Photocurrent at photo-electrode (powder solids were deposited onto the graphene working electrode) measured under intermittent UVA Hg lamp irradiation (150 W, $\lambda_{max}$ = 365 nm) in 0.1 M KOH.

To further investigate the relationship between the amount of exposed (001) facets and the photocatalytic activity of anatase TiO₂ samples, a terephthalic acid TA fluorescence probe was used to monitor the level of OH· radicals generated in the reaction system under UVA light irradiation (Figure 6). TA (fluorescence inactive compound) reacts with OH· radicals to form TAOH, therefore the fluorescence intensity of TAOH can be related to the level of OH· radicals produced by a particular catalyst sample. One should note that no oxidation of TAOH, which could influence the obtained data, took place during these measurements. As a result of the longer life-time of the charge carriers and lower degree of recombination of electron-hole pairs in the TiO₂-3 sample, the fluorescence signal of TAOH was stronger than the responses belonging to TiO₂-1 and TiO₂-2 solids. These observations are consistent with the results of the photocurrent measurements (Figure 5). Furthermore, this clearly indicates

that more electron-hole pairs are generated and more OH· radicals participate in the radical-chain reaction when the catalyst contains a larger amount of exposed (001) anatase facets. For illustration, the following production of OH· radicals was obtained in the given range of experimental conditions after 1 h of illumination with UVA light: 539 μmol/g$_{cat}$ (material TiO$_2$-1—the lowest amount of exposed (001) anatase facets), 572 μmol/g$_{cat}$ (material TiO$_2$-2), and 635 μmol/g$_{cat}$ (material TiO$_2$-3—the largest amount of exposed (001) anatase facets).

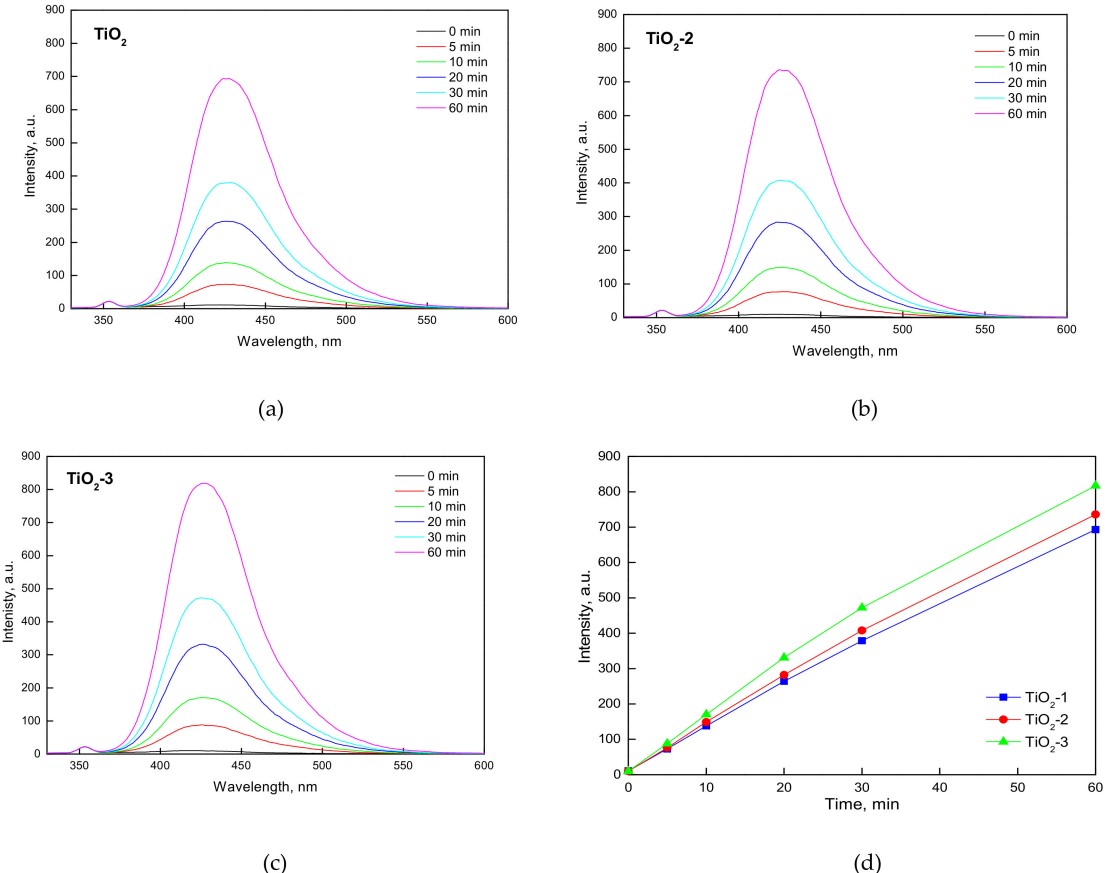

(a)　　　　　　　　　　　　　　　　　　　　　　　(b)

(c)　　　　　　　　　　　　　　　　　　　　　　　(d)

**Figure 6.** Fluorescence signal intensity of 2-hydroxyterephthalic acid (TAOH) vs. time over (**a**) TiO$_2$-1, (**b**) TiO$_2$-2, and (**c**) TiO$_2$-3 after irradiation with UVA light; (**d**) intensity of fluorescence signal of TAOH over TiO$_2$-1, TiO$_2$-2, and TiO$_2$-3 at λ = 425 nm as a function of time.

Based on the results of the photo-electrochemical and photo-luminescence measurements, the following decreasing order of the photocatalytic activity of the synthesized catalysts can be predicted: TiO$_2$-3 > TiO$_2$-2 > TiO$_2$-1. The predicted order was tested in the subsequent experiments of photocatalytic BPA degradation (Section 2.2).

### 2.1.3. TPD Analysis

The NH$_3$- and CO$_2$-TPD profiles recorded with all three TiO$_2$ materials used in the present study are depicted in Figure 7. The concentrations of acid and basic sites, shown as the total amount and the amount of strong sites (>350 °C for NH$_3$ and >450 °C for CO$_2$ desorption, respectively) are listed in Table 2. The surface acidity, both the total amount and concentration of strong acid sites, was about the same for TiO$_2$-1 and TiO$_2$-2 solids and increased significantly in the TiO$_2$-3 sample (factor of 1.6 to 1.9). In contrast, the total amount of basic sites changed little in the three samples while the amount of strong basic sites decreased significantly in the TiO$_2$-3 sample (factor of 3 to 4).

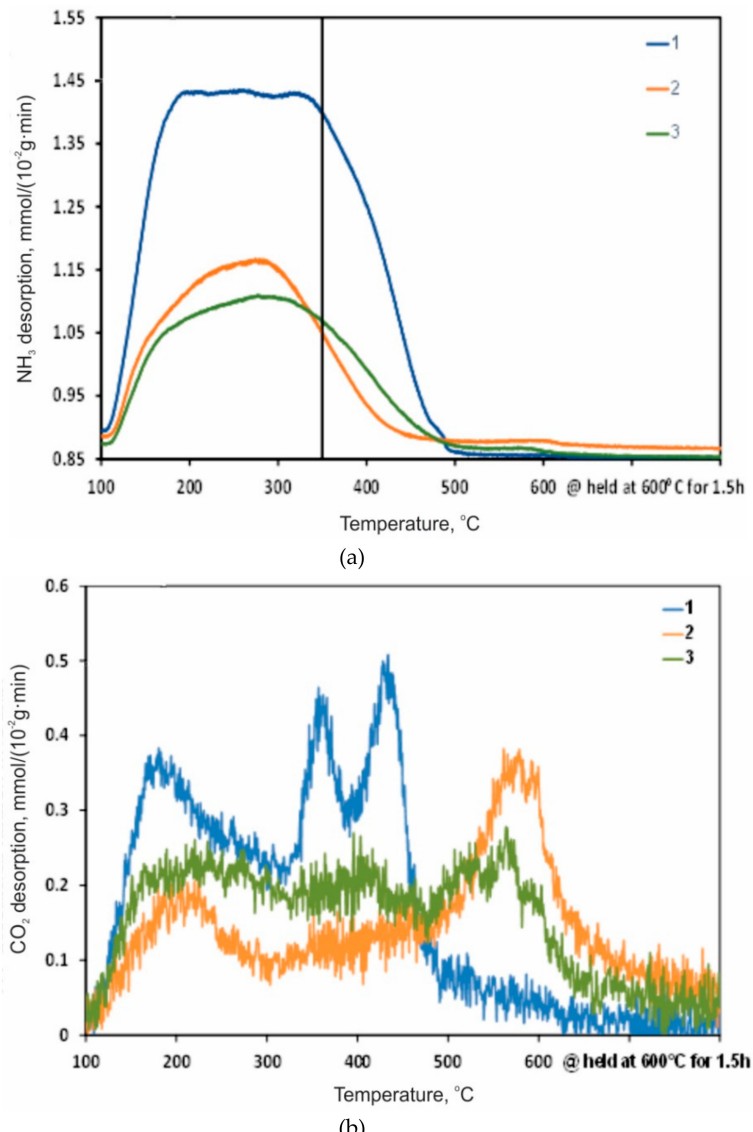

**Figure 7.** NH$_3$- and CO$_2$-TPD profiles recorded with TiO$_2$ materials used in the present study: (**a**) NH$_3$-TPD, (**b**) CO$_2$-TPD. Legend: 1—TiO$_2$-1, 2—TiO$_2$-2, 3—TiO$_2$-3.

**Table 2.** The concentrations of acid and basic sites, shown as the total amount and the amount of strong sites, at the surface of TiO$_2$ solids used in the present study according to NH$_3$- and CO$_2$-TPD data.

| Sample | Acidity (NH$_3$-TPD), mmol NH$_3$/g$_{cat}$ | | Basicity (CO$_2$-TPD), mmol CO$_2$/g$_{cat}$ | |
|---|---|---|---|---|
| | Total | Strong (>350 °C) | Total | Strong (>450 °C) |
| TiO$_2$-1 | 3.7 ± 0.4 | 1.1 ± 0.1 | 0.17 ± 0.02 | 0.06 ± 0.006 |
| TiO$_2$-2 | 3.8 ± 0.4 | 1.3 ± 0.1 | 0.14 ± 0.01 | 0.08 ± 0.008 |
| TiO$_2$-3 | 5.9 ± 0.6 | 2.1 ± 0.2 | 0.16 ± 0.02 | 0.02 ± 0.002 |

The TPD data provide evidence that changing the exposure of (001) facets in nanocrystalline anatase material does not affect its total surface basicity. The reduction of basic strength with increasing exposure of (001) nanocrystal facets (in contrast to (101) facets) may be attributed to the strained configuration of oxygen atoms, leading to destabilization of the 2p states on the surface oxygen ions. This is a result of the significantly higher angles of Ti-O-Ti bonds (146° (001) vs. 102° (101)) [9,35].

The role of Lewis acid sites mainly found at the anatase surface by spectroscopic methods [21,41] is determined by under-coordinated Ti(4+) ions [22]. The increase of surface acidity and strength of acid

sites with increasing exposure of (001) nanocrystals planes may be related to the higher concentration of most acidic under-coordinated Ti(4+) ions at these facets [35].

Anatase also displays Brønsted acidity that is related to the existence of surface hydroxyls [42]. These sites may also contribute to the increase of the total acidity and acid strength with the increase of the surface area represented by (001) facets. The protonation of anatase is attributed to the hydration of the bare or "vacuum pure" surface that proceeds according to different mechanisms at the (101) and (001) facets [9,43,44]. At the low-energetic (101) facet, molecular associative adsorption creates covalent bonds between under-coordinated surface titanium atoms and oxygen from the water molecule: $H_2O$ $\cdots$ Ti[ ]. At the high-energetic (001) facet, the two-coordinated bridging oxygen atoms at the surface form strong hydrogen bonds with a water-hydrogen atom that induces dissociation of an adsorbed water molecule, creating surface titanols (Ti-OH): Ti-O-Ti + $H_2O$ → 2Ti-OH.

Due to the different chemistry of the hydration layer formed at the surface of (101) and (001) facets of anatase nanocrystals, molecularly adsorbed water and titanols, respectively, its further interaction with water molecules leads to a different surface chemistry [27]. Weak hydrogen bonding at (101) facets forms a layer analogous to bulk water, while strong hydrogen bonding with surface hydroxyls serving as hydrogen bond donors at (001) facets creates a layer containing acidic protons [45]. A similarly strong interaction between surface silanols and water molecules was found in [45]. Increasing the contribution of (001) facets to the total surface area in the $TiO_2$-3 sample increased the concentration of surface titanols and acidic surface protons as indicated by MAS ss NMR data.

### 2.1.4. NMR Examination

[1]H MAS ssNMR spectra of the materials, $TiO_2$-1, $TiO_2$-2, and $TiO_2$-3 (Figure 8a), revealed two distinct differences. The concentration of titanol sites resonating between 0 and 1.5 ppm increased, and the signals at ~5.5 ppm broadened significantly in the $TiO_2$-3 sample. The Ti-OH group concentration obtained by deconvolution of the NMR signals is vanishingly small in the $TiO_2$-1 sample, and increased to 4% of the total intensity in the $TiO_2$-2 solid. The high exposure of the (001) facets in the $TiO_2$-3 sample increased the Ti-OH group concentration to 15%. The increased concentration of titanols observed by NMR analysis is in good agreement with the FTIR data. This may be attributed to dissociative adsorption of water molecules at the (001) facets of anatase nanocrystals versus molecular water adsorption at low-energetic (101) facets. As shown by DFT calculations, the high density of surface under-coordinated titanium ions and very strained configuration of all surface atoms at (001) anatase facets cause an increased surface reactivity favoring water dissociation, with the formation of Ti-OH groups [9]. The signals of isolated TiOH titanol groups mainly appear at 2 to 3 ppm, while acidic bridging hydroxyl groups appear at 6.5 to 7 ppm [46,47]. Signals at around 0 ppm have also been observed and attributed to internal defect sites containing terminal hydroxyl groups. Interestingly, shifts in Figure 8a appear at somewhat lower values, between 0.8 and 1.4 ppm for TiOH groups, and the largest signal for acidic –OH groups was at 5.4 to 5.7 ppm.

One of the major contributions to the enhanced reactivity of the (001) surface is the chemisorption of water on a surface with an increased concentration of titanol groups. The acidic –OH signal in proximity to any of those titanol groups was examined by performing a 2D [1]H-[1]H correlation experiment. The spectrum plotted in Figure 8b clearly shows the transfer of magnetization from the OH signals to the two main TiOH sites of the $TiO_2$-3 sample, indicating a distance that is lower than 1 nm between those species. The appearance of the correlation peak and the broadening of the signal are probably a result of the high concentration and low dynamics of those species. The correct interpretation of the 2D spectrum could be either water is bound to titanols or a mixture of bridging hydroxyls and titanols. The latter is less likely due to the higher shift of bridging hydroxyls (6.5–7 ppm) compared to the measured 5.4 to 5.7 ppm and it does not explain the broadening of only this signal. This is also supported by a study of anatase nanotubes showing similar shifts of bound water [48]. Therefore, it is most likely that the (001) surface is covered with acidic rigid water molecules anchored on the titanols by hydrogen bonds as indicated by the shifting to higher ppm with the increasing of the

contribution of (001) facets to the total surface area. Their protons have a higher acidity compared with water molecularly adsorbed at the (001) facets. The narrower signals of water in the $TiO_2$-2 sample correspond to isolated water signals, probably bound directly to titania surface atoms at (101) facets.

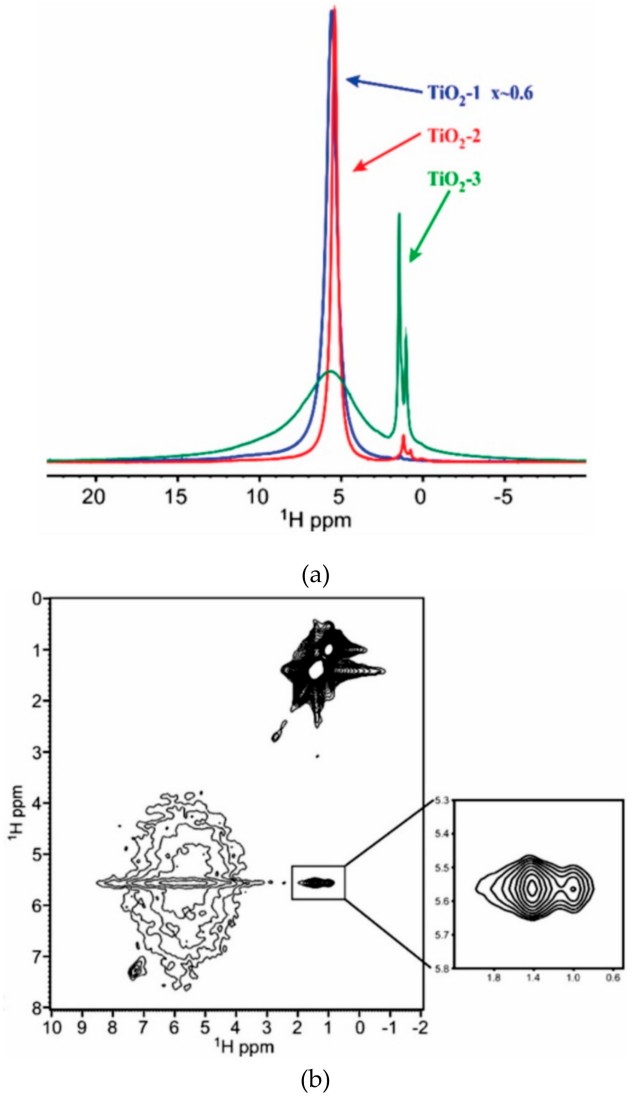

**Figure 8.** (**a**) $^1$H MAS ssNMR spectra of $TiO_2$-1 (blue), $TiO_2$-2 (red), and $TiO_2$-3 (green) materials; (**b**) 2D $^1$H-$^1$H correlation spectrum of $TiO_2$-3 sample. Main $TiO_2$-2 signals are at 0.8, 1.2, and 5.4 ppm. Main $TiO_2$-3 signals are at 1.0, 1.5, and 5.7 ppm, with a clear, but small contribution also at 1.3 ppm.

$^{49}$Ti spectra under MAS NMR are difficult to measure due to its high spin (S = 7/2), low abundance (5.4%), and low gyromagnetic ratio (5.6% of that of $^1$H). Ti MAS spectra of a commercial anatase sample revealed a single cylindrically symmetric site with a quadrupole coupling constant of 4.6 MHz [49]. In a different study of various $BaTiO_3$ complexes, a finely grained $TiO_2$ anatase [50] spectrum revealed a broader and more featureless line, which upon fitting to a single site yielded ~ 5 MHz. FAM-enhanced spin-echo Ti spectra depicted in Figure 9 for $TiO_2$-2 and $TiO_2$-3 samples yielded broadened features that could not be fitted with a single site. A good fit to two sites yielded quadrupolar coupling constant values of 2.7 MHz (η = 0.6) and 3.7 MHz (η = 0.46) for the $TiO_2$-2 sample, and larger values for the $TiO_2$-3 solid with increased exposure of (001) facets, i.e., 3.2 MHz (η = 0.43) and 4.2 MHz (η = 0.15). While it is possible that a larger distribution of sites exists, in particular due to the increased number and exposure of surface species, it reflects differences in the two structures, having a five-fold alteration in the exposure of (001) anatase facets. It demonstrates the variability and enrichment of Ti site types in

anatase nanocrystals with increased exposure of anatase (001) facets. This may be a result of increasing the amount of under-coordinated Ti-atoms bonded to –OH groups.

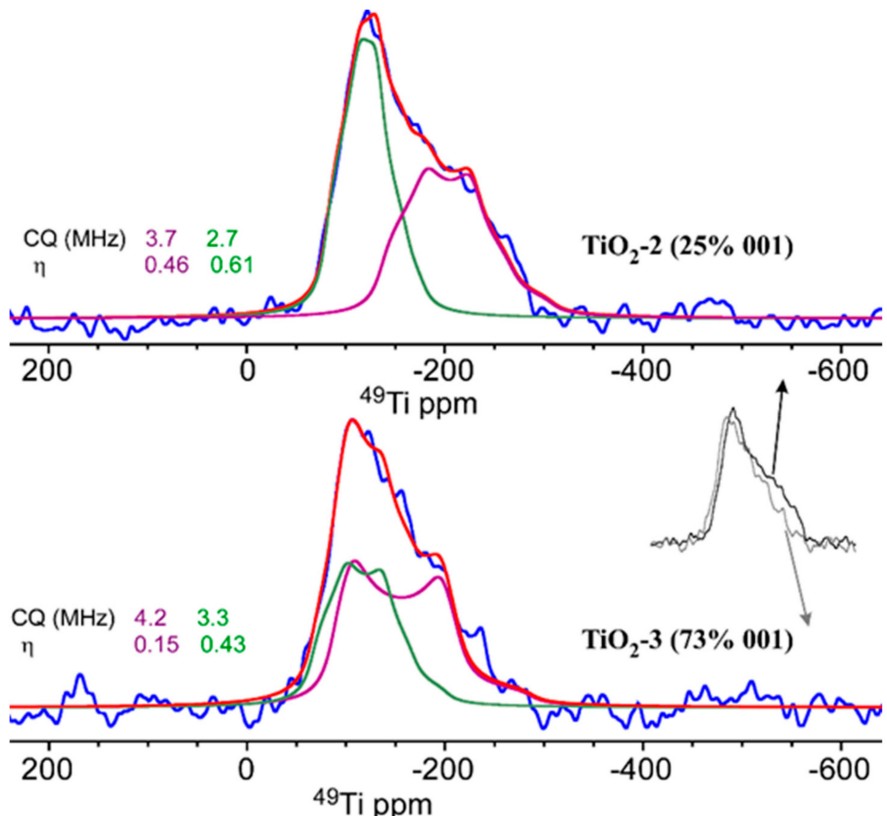

**Figure 9.** $^{49}$Ti FAM-enhanced MAS ssNMR spectra of TiO$_2$-2 and TiO$_2$-3 samples with their deconvolution to a minimal set of two sites. Overlay of the spectra shows the variations between the two materials.

## 2.2. Photocatalytic BPA Oxidation

Figure 10 shows the BPA concentration-time degradation curves obtained in the presence of synthesized materials. To determine the extent of BPA adsorption on the catalyst surface, the experiments were conducted for 30 min without illumination (dark phase). The obtained results show that the decrease of BPA concentration in the dark phase was negligible, thus its adsorption could be neglected. After 30 min, the BPA solution containing dispersed catalyst powder was illuminated with the same UVA lamp as used in the photocurrent and TAOH photo-luminescence measurements. The measured BPA relative concentrations vs. time dependencies were found to be reproducible with an error of BPA conversions measured within ±1%. No leaching of TiO$_2$ into the liquid phase was detected in any of the runs performed. Furthermore, the photolytic degradation of BPA under UVA illumination was almost negligible (Figure 10, black line). The initial stages (10 min) of each reaction run display a decreasing order of catalyst activity for BPA degradation: TiO$_2$-3 > TiO$_2$-2 > TiO$_2$-1 (the initial reaction rates were found to be 4.8, 2.9, and 2.4 mg/(g$_{cat}$·min) for samples of TiO$_2$-3, TiO$_2$-2, and TiO$_2$-1, respectively). The observed order of photocatalytic activity correlates very well with the findings of the photocurrent and photo-luminescence measurements. It further shows that with the increasing amount of exposed (001) anatase facets, the catalytic activity of samples increases. Overall, we can conclude that initially BPA oxidation occurs on active centers located at TiO$_2$ nanoparticles along long-dimensional crystal planes (and not on edges).

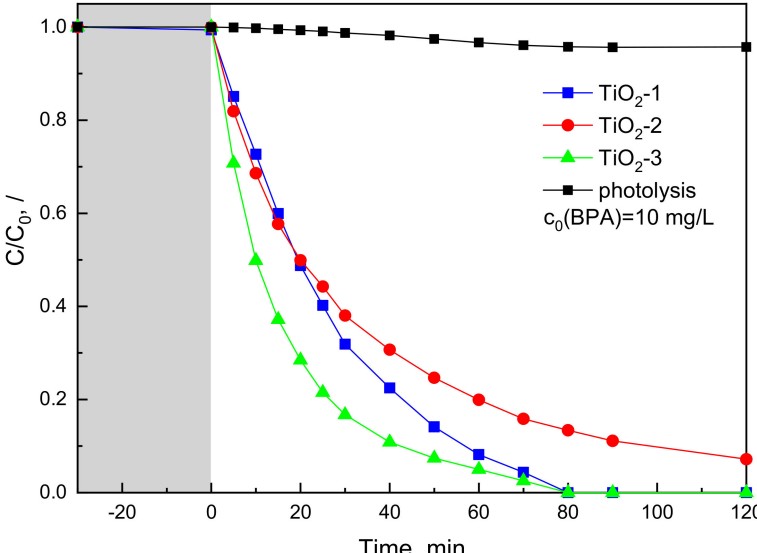

**Figure 10.** Photocatalytic degradation of aqueous BPA solution in the presence and absence of prepared catalysts irradiated with a UVA Hg lamp (150 W, $\lambda_{max}$ = 365 nm). The catalyst concentration used in the performed experiments was 125 mg/L.

However, with a prolongation of the BPA degradation runs, the observed order is not valid any more. One can see in Figure 10 that the $TiO_2$-1 and $TiO_2$-3 solids achieved 100% BPA removal after 80 min in comparison to the $TiO_2$-2 sample, where the relative amount of remaining BPA in the aqueous solution was still around 7% after 120 min of illumination with UVA light. Table 3 lists TOC removal data as well as values about the extent of total mineralization of the parent organic matter ($TOC_m$) obtained after 120 min of UVA irradiation of aqueous BPA solution. Taking into account both the activity data (Figure 10) as well as the extent of BPA mineralization and its reaction intermediates (Table 3), the photocatalytic efficiency of the examined samples at the end of the oxidation runs decreases in the following order: $TiO_2$-1 > $TiO_2$-3 > $TiO_2$-2. Based on the TOC results and the shape of the concentration-time curves of the BPA degradation runs performed over the catalysts, $TiO_2$-2 and $TiO_2$-3 (Figure 10), we assume that the deposition of BPA degradation intermediates onto the catalyst surface could play an important role in the observed catalytic performance. Furthermore, the results of the CHNS measurements performed on fresh ($TC_{fresh}$) and spent ($TC_{used}$) catalyst samples (Table 3) revealed that with the increasing amount of exposed (001) anatase facets, the amounts of carbonaceous deposits during the BPA degradation runs increased too. The lowest extent of BPA mineralization (i.e., 24%) was obtained in the presence of the $TiO_2$-3 sample. This was also confirmed by the thermogravimetric analysis of spent catalysts (Figure 11), where it was found that the spent $TiO_2$-3 sample exhibited the highest loss of mass. The comparison of the results of CHNS and TGA-TPO measurements indicates that deposits on the surface of spent $TiO_2$-1, $TiO_2$-2, and $TiO_2$-3 catalysts contain 16%, 26%, and 49% of the initial carbon, respectively. FTIR-ATR spectra of spent catalyst samples (Figure 4) display various vibrations belonging to -$CH_2$ and -$CH_3$ groups (peaks at 1280, 2855, 2926, and 2962 $cm^{-1}$). Vibrations of bands of aromatic carbon species (C-H vibrations above 3000 $cm^{-1}$) could not be detected. Deposition of BPA degradation intermediates onto the surface of active (001) facets drastically decreases the photocatalytic activity of $TiO_2$-2 material in the later stage of the reaction course as depicted in Figure 10. Since the initial BPA concentration in the solution is 10 mg/L and the $TiO_2$-3 solid has a much higher surface of exposed (001) facets than the $TiO_2$-2 sample (Table 1), the amount of BPA degradation intermediates was not sufficient to block the catalytic active (001) facets in the $TiO_2$-3 material. As a result, BPA degradation conducted in the presence of the $TiO_2$-3 sample was completed as opposed to the $TiO_2$-2 solid.

**Table 3.** Carbon content accumulated on the surface of catalysts before ($TC_{fresh}$) and after ($TC_{used}$) photocatalytic BPA degradation. TOC removed from the liquid phase ($TOC_R$) represents a sum of true TOC conversion—mineralization ($TOC_m$) and TOC accumulation on the catalyst surface as adsorbed and not soluble BPA oxidation intermediates ($TOC_{ads}$). TOC content remained in water as soluble BPA oxidation intermediates is designated as $TOC_{Resid}$.

| Sample | $TOC_R$ | $TC_{fresh}$ | $TC_{used}$ | $TC_{used} - TC_{fresh}$ | $TOC_m$ | $TOC_{ads}$ | $TOC_{Resid}$ |
|---|---|---|---|---|---|---|---|
| | (%) | | (wt. %) | | | (%) | |
| $TiO_2$-1 | 89 | 0.28 | 0.99 | 0.71 | 77 | 12 | 11 |
| $TiO_2$-2 | 65 | 0.28 | 1.63 | 1.35 | 43 | 22 | 35 |
| $TiO_2$-3 | 72 | 0.18 | 3.09 | 2.91 | 24 | 48 | 28 |

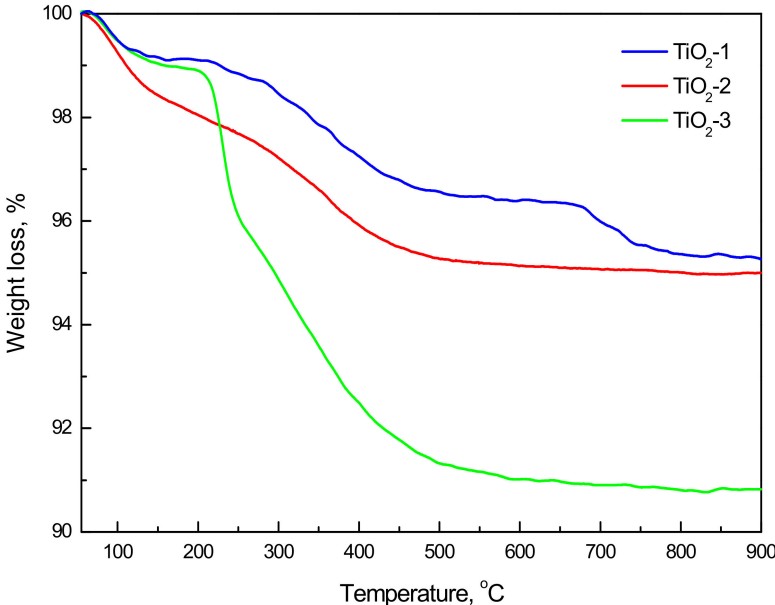

**Figure 11.** Thermogravimetric TGA-TPO analysis of catalyst samples after use in BPA degradation runs.

Comparison of the efficiency of the investigated $TiO_2$ materials in oxidative photocatalytic mineralization of water containing BPA shows that the processes taking place in the system may be presented by the scheme shown in Figure 12. The BPA mineralization proceeds to $CO_2$ through the formation of degradation intermediates (DI) that are (i) partially converted to $CO_2$, (ii) partially remain in the solution ($TOC_{Resid}$, Table 3), or (iii) adsorb to the catalyst surface ($TOC_{ads}$, Table 3), reacting with acidic –OH groups.

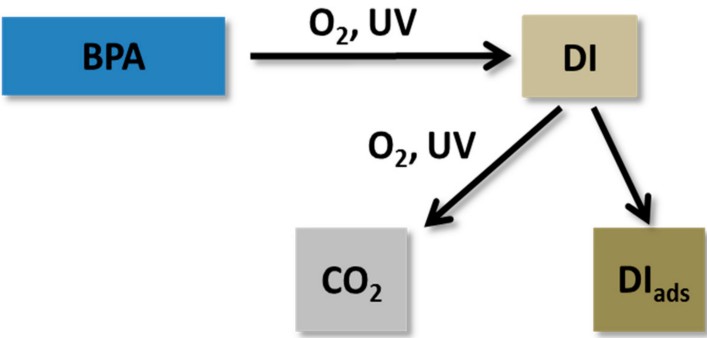

**Figure 12.** Scheme of several BPA transformation routes (DI—degradation intermediates, $DI_{ads}$—adsorbed degradation intermediates).

## 3. Materials and Methods

### 3.1. Catalyst Preparation

The commercial high surface area $TiO_2$ material with anatase structure (sample $TiO_2$-1) was purchased from Saint-Gobain NorPro Co, Stow, OH, USA. (anatase material XT25376). The $TiO_2$ sample denoted as $TiO_2$-2 was prepared by means of a solvo-thermal method. In total, 1 mL of titanium (IV) isopropoxide (>98.0%, Acros, Geel, Belgium) was dropped slowly into a beaker containing 28 mL of isopropyl alcohol (>99.5%, Daejung, Sheung-city, Korea) at room temperature under stirring. The resulting mixture was transferred into a Teflon-lined stainless autoclave (35 mL capacity). The autoclave was sealed and maintained at T = 200 °C for 24 h. The system was then cooled to ambient temperature. The final solid product was collected by centrifugation, washed thoroughly with ethanol, and dried overnight at 70 °C. The product was calcined at 400 °C for 2 h with a heating rate of 1 °C min$^{-1}$. $TiO_2$ nanosheets (sample $TiO_2$-3) were prepared by means of a hydrothermal method in the presence of fluorine. In total, 5 mL of titanium (IV) isopropoxide (>98.0%, Acros, Geel, Belgium) and 0.8 mL of hydrofluoric acid (48% solution, Sigma Aldrich) were mixed in a Teflon vial and stirred for 5 min at room temperature. The resulting solution was transferred to a Teflon-lined stainless autoclave (35 mL capacity). The autoclave was sealed and maintained at 180 °C for 24 h. After hydrothermal reaction, the system was allowed to cool to ambient temperature. The solid product was collected by centrifugation, washed thoroughly with ethanol and distilled water, and dried overnight at 80 °C. For removal of residual fluoride ($TiOF_2$ phase), the as-prepared $TiO_2$ material was washed first with a diluted NaOH solution (0.1 M, Frutarom, Haifa, Israel), then several times with distilled water and finally dried overnight at 80 °C.

### 3.2. Catalyst Characterization

FTIR spectrometer from Perkin Elmer (Waltham, MA, USA, model Frontier), equipped with the GladiATR Vision$^{TM}$ accessory (PIKE Technologies) containing a diamond crystal, was used to record the FTIR-ATR spectra of fresh and used catalyst samples. The spectral range of the obtained spectra (average of 64 scans) was between 4000 and 450 cm$^{-1}$ with a resolution of 4 cm$^{-1}$.

A Metrohm Autolab (Ultrecht, The Netherlands) PGSTAT30 potentiostat/galvanostat and a three-electrode electrochemical cell were used to evaluate the photo response characteristics of the synthesized materials under UVA light illumination (UVA Hg lamp, 150 W, $\lambda_{max}$ = 365 nm). A drop (10 μL) of a catalyst-ethanol suspension (5 mg of catalyst dispersed in 1 mL of absolute ethanol (Sigma Aldrich)) was dropped onto the surface of a screen-printed electrode (DropSens, Asturias, Spain, model DRP-150) and dried overnight at room temperature to prepare the working electrode. A calomel electrode from HANNA Instruments (Woonsocket, RI, USA, model HI5412) presented the reference electrode and a platinum sheet was used as the counter electrode. The electrolyte was an aqueous solution of KOH (0.1 M, Carlo Erba, Barcelona, Spain).

The amount of carbon accumulated on the catalyst surface during the photocatalytic oxidation runs was determined by measuring the carbon content of fresh ($TC_{fresh}$) and spent ($TC_{used}$) catalysts with a Perkin Elmer CHNS analyzer (Waltham, MA, USA, model 2400 Series II).

Used catalyst samples were thermogravimetrically analyzed (TGA-TPO) by means of Perkin Elmer STA 6000 instrument (Waltham, MA, USA). The measurements were performed in air utilizing a temperature ramp of 20 °C/min.

To measure the level of OH· radical formation, 5 mg of a photo-catalyst were suspended in 50 mL of an aqueous solution containing 20 mM NaOH (Merck) and 6 mM terephthalic acid (TA, Alfa Aesar, Haverhill, MA, USA). The suspension was stirred in the dark for 20 min before illumination with a UVA Hg lamp (150 W, $\lambda_{max}$ = 365 nm). During the illumination period, aqueous-phase samples were withdrawn in different time intervals and immediately filtered through a 0.2 μm membrane filter. The samples were then analyzed by recording the fluorescence signal of the generated 2-hydroxyterephthalic acid (TAOH) using a UV/Vis fluorescence spectrophotometer (Perkin Elmer,

Waltham, MA, USA, model LS 55). The wavelength of the excitation light was 315 nm, and the scanning speed was 600 nm min$^{-1}$. The widths of the excitation slit and the emission slit were both set to 5 nm. To quantitatively determine the concentration of photo-generated TAOH (or OH·), the photo-luminescence (PL) intensity of the standard compound, TAOH (Aldrich, Steinheim, Germany, 97%, p.a.), with various concentrations in 20 mM NaOH (Merck) aqueous solution was measured; the PL calibration curve used for quantification purposes was obtained by plotting the PL intensity measured at λ = 425 nm as a function of the TAOH concentration (Figure S1, see Supplementary Information).

The XRD patterns of solids were obtained using a PANalytical Empyrean Powder Diffractometer (Almero, The Netherlands) equipped with a position sensitive detector, X'Celerator, fitted with a graphite monochromator, at 40 kV and 30 mA. Software developed by Crystal Logic was used. The crystallographic phases were identified via a SBDE ZDS computer search/match program joined with the ICDD library. The average crystal sizes of the $TiO_2$ anatase phase were determined in two crystallographic directions—thickness and width—derived from the Scherrer equation: h = Kλ/(B$^2$ − β$^2$)$^{0.5}$·cos(2θ/2), where K = 1.000—shape factor, λ = 0.154 nm, β—instrumental broadening correction, and B—reflection broadening at corresponding 2θ. The normal crystal size was a result of an averaging of the results obtained for wide ((105), 2θ = 53.9°; (116), 2θ = 68.6°) and narrow ((200), 2θ = 47.9°; (220), 2θ = 70.1°) reflections of the anatase XRD patterns (ICDD card 21-1272).

The shape of $TiO_2$ nanocrystals was investigated with a field-emission analytical transmission electron microscope (JEOL, Tokyo, Japan, JEM-2100F), and an accelerating voltage of 200 kV was applied. The materials were examined on carbon coated copper grids after depositing a drop of ethanol suspension of the solid titania samples and drying in air.

The texture parameters—surface area, total pore volume, and average pore size of $TiO_2$ materials—were derived from $N_2$ adsorption–desorption isotherms using conventional BET and BJH methods. The isotherms were obtained on a NOVA3200e (Quantachrome, Boynton Beach, FL, USA) instrument at the temperature of liquid nitrogen after their outgassing under vacuum for 2 h at 100 °C.

The $NH_3$-TPD and $CO_2$-TPD profiles were obtained with an AutoChem II 2920 analyzer (Micromeritics, Norcross, GA, USA) equipped with a mass spectrometer (Cirrus 2, MKS, Andover, MA, USA). The TPD procedures were conducted as follows: Treatment in He flow of 25 mL/min for 1 h at 150 °C for dehydration, then cooling to 40 °C, saturation for 1 h with $CO_2$ or $NH_3$ under 25 mL/min flow of 5% $CO_2$/He for $CO_2$-TPD, or 5% $NH_3$/He for $NH_3$-TPD. The saturated samples were heated to 100 °C under He flow to remove weakly adsorbed gases, heated (5 °C/min) to 600 °C, and kept at this temperature for 1.5 h in He flow while recording the desorption profiles. Calibration of the TCD signal intensity with $CO_2$/He and $NH_3$/He mixtures of different compositions allowed the values of mmols/g$_{cat}$ of $CO_2$ or $NH_3$ based on recorded TPD profiles to be derived.

Magic angle spinning (MAS) solid-state NMR (ssNMR) experiments were performed on a Bruker (Billerica, MA, USA) Avance-III spectrometer operating at $^1$H and $^{49}$Ti frequencies of 400.1 and 22.6 MHz, respectively. $^1$H spectra were externally referenced to Adamantane at 1.8 ppm, and $^{49}$Ti spectra to $SrTiO_3$ at 0 ppm. $SrTiO_3$ was used to calibrate $^{49}$Ti pulses. Bloch-decay $^1$H MAS spectra were acquired with 64 scans, recycle delays of 2 to 5 s, and at a spinning frequency of 12 kHz. The acquisition times were 40 ms and the data were apodized with a slight line broadening of 1 Hz. $^1$H-$^1$H correlation spectra were acquired on a $TiO_2$-3 sample using a mixing time of 10 ms, acquisition times of 100 and 20 ms and processed with 50 Hz line broadening (direct) and a cosine-squared function (indirect). $^{49}$Ti spectra were obtained by initially enhancing the central transition (CT) population by fast amplitude modulation (FAM) pulses (or RAPT: Rotor assisted population transfer) [51,52] followed immediately by a Hahn-echo using selective CT pulses with a radio-frequency power of 31 kHz. This approach allows for significant signal enhancement of $^{49}$Ti, which is a spin-7/2 with a low abundance (5.4%), and enabled the acquisition of the spectra using 20 to 30 thousand scans employing a recycle delay of 2 s. For signal enhancement, we used a FAM cycle of 5.6 μs with equal segments of

1.4 μs ({*x* pulse, delay, $\bar{x}$ -pulse, delay} × *n*) for a total of 84 μs (*n* = 15, ~one rotor period), and a power of 77 kHz. Deconvolution of $^1$H spectra was performed with dmfit [53], and deconvolution of $^{49}$Ti spectrum to two sites was performed with Topspin v3.5.

### 3.3. Photocatalytic Oxidation Experiments

A batch slurry reactor (Lenz, Wertheim, Germany, model LF60, 250 mL) was used to perform photocatalytic experiments at 20 °C (Julabo, Seelbach, Germany, model F25/ME) and atmospheric pressure. During the entire oxidation run, the aqueous solution (ultrapure water, 18.2 MΩ cm) of bisphenol A (BPA, $c_0$ = 10 mg/L, Aldrich) was purged with purified air (45 L/h) and magnetically stirred (600 rpm). Ultrasonication was employed to suspend the catalyst (125 mg/L) before adding to the BPA solution. The sorption process equilibrium was established during the 30 min in which the aqueous suspension was kept in the dark ("dark" period). After that, the suspension was illuminated with a UVA Hg lamp (Philips, Amsterdam, The Netherlands, 150 W, maximum at λ = 365 nm) positioned in a water-cooled quartz jacket immersed vertically in the middle of the batch slurry reactor.

### 3.4. Analysis of End-Product Solutions

During the photocatalytic runs, representative 1.5 mL samples were withdrawn in 5 to 30 min intervals from the reactor and immediately filtered through a 0.2 μm membrane filter before being analyzed with an HPLC instrument (Thermo Scientific, Waltham, MA, USA, model Spectra) to determine temporal BPA conversions. The isocratic analytical mode using a 100 mm × 4.6 mm BDS Hypersil C18 2.4 μm column thermostated at 30 °C was used to perform HPLC measurements. The flow rate of the mobile phase (70% methanol (Merck, Darmstadt, Germany) and 30% ultrapure water) was 0.5 mL/min, and UV detection was conducted at λ = 210 nm. A Teledyne Tekmar total organic carbon (TOC) analyzer (model Torch, Mason, OH, USA), applying a high-temperature (750 °C) catalytic oxidation (HTCO) method, was used to measure the total organic carbon (TOC) content in the solution at the beginning and end of the photocatalytic experiments. This was applied to determine the total amount of removed organic substances from the liquid phase ($TOC_R$) as well as the mineralization level (i.e., the extent of parent organic matter transformed to carbon dioxide, $TOC_m$). The reported TOC values are an average of three repetitions, where the observed error was within ±1%.

## 4. Conclusions

One way to improve the photocatalytic activity of anatase $TiO_2$ is to increase the amount of exposed (001) facets, which are much more reactive in the dissociative chemisorption of water than (101) facets. Three samples of anatase $TiO_2$, with the exposure of (001) facets increasing from 12% for $TiO_2$-1 to 38% for $TiO_2$-2 and 63% for $TiO_2$-3, were employed in photocatalytic degradation of water dissolved bisphenol A (BPA). TDP measurements revealed a similar surface acidity of $TiO_2$-1 and $TiO_2$-2 samples and higher values of both total and strong acidity (factor of 1.6 to 1.9) for $TiO_2$-3 solid. The increase in the concentration of Ti-OH groups with an increase in the amount of exposed (001) facets was detected by FTIR measurements and quantified by the deconvolution of NMR signals. The amount of photo-generated photocurrent and the intensity in fluorescence probing under UVA light irradiation of solids increased with an increasing amount of exposed (001) facets. Based on these findings, the activity of the tested solids in photocatalytic liquid-phase oxidative degradation of BPA was anticipated to be $TiO_2$-3 > $TiO_2$-2 > $TiO_2$-1, as indeed measured during the initial stage of BPA degradation.

The increase of the surface area of exposed (001) facets yielded a higher amount of acidic –OH groups that adsorb degradation intermediates on the catalyst surface. Consequently, catalytic materials with high exposure of (001) facets display decreased photocatalytic activity at the later stage of a reaction. As a result, the global extent of BPA mineralization decreased with increased exposure of (001) facets: $TiO_2$-1 > $TiO_2$-2 > $TiO_2$-3. The decrease of catalytic activity caused by the deposition of degradation intermediates was especially prominent for the $TiO_2$-2 sample, which is due to its lower

surface area compared to the other samples. Taking full advantage of the higher photocatalytic activity of TiO$_2$ materials with high exposure of (001) facets requires modification of catalyst surface chemistry to reduce the adsorption of partially oxidized reaction intermediates.

**Supplementary Materials:** The following are available online at http://www.mdpi.com/2073-4344/9/5/447/s1, Figure S1: PL calibration curve obtained by plotting the PL intensity measured at λ = 425 nm as a function of 2-hydroxyterephthalic acid (TAOH) concentration.

**Author Contributions:** Conceptualization, A.P., M.H. and M.F.; validation, G.Ž., M.L., M.F., A.G. and A.H.; G.Ž., M.L., M.F., A.G. and A.H.; resources, G.Ž., M.L., M.F., A.G. and A.H.; data curation, G.Ž., A.H., M.F. and A.G.; writing—original draft preparation, G.Ž.; writing—review and editing, A.P., M.H. and M.L.; visualization, G.Ž., M.F., M.L., A.H. and A.G.; supervision, A.P. and M.H.; project administration, A.P. and M.H.; funding acquisition, A.P. and M.F.

**Funding:** This research was funded by the Slovenian Research Agency (research core funding No. P2-0150) and the Ben Gurion University—University of Chicago Institute for Molecular Engineering—Argonne National Laboratory Collaborative Program on Molecular Engineering of Water Resources.

**Conflicts of Interest:** The authors declare no conflict of interest.

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
