# Peer review of "Effect of Surface Chemistry and Crystallographic Parameters of TiO2 Anatase Nanocrystals on Photocatalytic Degradation of Bisphenol A"

_catalysts, doi:10.3390/catal9050447_

Round 1

Reviewer 1 Report

General Points:

The experiments performed are feasible and the results reported seem possible

The techniques used are up to date and have been handled with care

The degradation of BPA is an important ecological problem to address since it disrupts the the endocrine system of humans. Therefore innovative more effective catalysts/methods should be investigated to detoxify water bodies containing this pollutant.

The authors provide evidence in their work for  exposed (001) TiO2 facets  in TiO2 materials led to a larger surface area in the TiO2 materials and these factors were beneficial during the  light induced BPA-degradation.TThe predominantly Bronsted acidity for the anatase with (001) facets is documented by several techniques: FTIR, TPD, MAS,TPD, TGA, HPlC, TOC and NMR.

The authors characterize the surface of diverse TiO2 materials by XRD and TEM. The Ms provides the evidence that a larger amount of OH-radicals are produced when a high amount of exposed (001) facets are present.They have used the 2-hydroxy-terephthalic acid fluorescence to monitor quantitatively the OH-radical amounts generated under light by the (001) facets of anatase materials and correlate the findings to the amount of BPA-degradation.

Specific Points:

Mention in the abstract the time of BPA photo-degradation by the best anatase catalyst and the initial concentration of the BPA used.

Add in the legend in Figure 10 the initial concentration of BPA applied at time zero.

Shorten the text described in lines 336 to 3922. A detailed discussion of  TiO2-1,TiO2-2 and TiO2-3 is not warranted due to the closeness of the traces and the possible experimental error involved.The text should be shortened by 40%. A  more precise write-up with the main results leaving aside many speculations is required for this material.

Section 3.4. What product/ transient intermediate (s) do you analyze by HPLC? The full degradation of BPA is shown in Figure 10. Therefore only C-containing intermediates can be reported but I did not find the pertinent information. Explain.

Reviewer 2 Report

In this manuscript, the authors present a systematic investigation of the role of (001) facets on the photocatalytic reactivity of titania nanocrystals for degradation of BPA. They prepare three different sets of samples with varying population of facets, the structure of which are thorough characterized.  Interface acidity and other relevant properties are explored to determine the potential mechanistic underpinnings of the observed differences in reactivity, demonstrating that (001) facets’ greater acidity (and the correlated enhanced sorption of water) is the primary factor.  While the literature on TiO2 photocatalysis is large, this work adds a novel contribution that furthers the understanding and therefore warrants publication.  Just a few small revisions are recommended:

- The authors mention the drawback of TiO2 related to its large bandgap (and associated requirement for UV activation), but they do not mention important strategies for overcoming this limitation.  One recent example is nitrogen doping of the TiO2 to create mid-gap states and enable visible light activation (e.g. Adv. Sust. Sys. 1 (2017) 1600041).

- Tables 1 and 2 need to have standard deviation or other +/- values for all values.

- In Figure 5, the trace associated with TiO2-3 clearly has more noise and other unexplained featured during illumination.  The authors should discuss the origin of these features in the discussion.

- Often, photocatalytic degradation curves are approximately linear with time.  The data in Figure 10 are non-linear.  Can the authors provide some discussion of the kinetics involved?
